# In Silico Modeling the Impact of Cartilage Stiffness on Bone Tissue Stress

**Vidmantas Alekna [1], Oleg Ardatov [2,*], Jelena Selivonec [3] and Olga Chabarova [3]**

[1] Faculty of Medicine, Vilnius University, LT-03101 Vilnius, Lithuania; vidmantas.alekna@mf.vu.lt
[2] Faculty of Mechanics, Vilnius Gediminas Technical University, LT-10223 Vilnius, Lithuania
[3] Faculty of Civil Engineering, Vilnius Gediminas Technical University, LT-10223 Vilnius, Lithuania; jelena.selivonec@vilniustech.lt (J.S.); olga.chabarova@vilniustech.lt (O.C.)
[*] Correspondence: oleg.ardatov@vilniustech.lt

**Featured Application: The findings of this investigation, along with the modeling approach presented, may prove beneficial in the field of trauma biomechanics and medical diagnosis, providing a more thorough evaluation of a patient's health status.**

**Abstract:** The knee joint is a complex biomechanical subsystem, modeling of which can reveal a deeper understanding of the processes occurring within it. The purpose of this study is to examine the stress alteration in bone based on mechanical properties of cartilage. To achieve this, a numerical model of the knee joint was developed and tested under different displacement values. The mechanical behavior of the model was represented by considering the hyperelastic properties of soft tissues, along with the verification of trabecular structure of bones, resulting in a more realistic mechanical depiction of the biological subsystem. The results showed that as the stiffness of the cartilage increased; the distribution of stresses in the bone became uneven; and stress concentrators dispersed over articular surface, while in the case of mild cartilage no stress concentrators were expressed. The proposed modeling approach allows the adaptation of patient-specific data in order to predict the outcomes of tissue diseases. The obtained results allow us to state that taking into account the non-linear properties of soft tissues is extremely important for assessing the stress state of the entire biological subsystem. The main difficulty, however, is the lack of data regarding the mechanical behavior of tissues in certain diseases.

**Keywords:** biomechanics; cartilage; femur; hyperelasticity; meniscus; numerical modeling; tibia; trabeculae

## 1. Introduction

In silico modeling is a valuable tool for investigating the mechanical behavior of biological systems and exploring treatment options for various diseases [1]. Due to advancements in computational power and imaging technology, in silico modeling has found widespread application in the field of orthopedics [2], specifically in knee joint biomechanics, including the modeling of knee joint diseases such as osteoarthritis (OA) [3]. Patient-specific finite element models, simulating clinical changes in the cartilage, can predict how tissue deterioration affects bone mechanics and patient outcomes [4]. Recent studies have shown that in cases of OA and other degenerative soft tissue diseases, the stiffness of cartilage decreases [5,6]. Deterioration of cartilage, in turn, disrupts the natural functionality of the entire joint, in particular, having a negative impact on bone tissue, which wears out more intensively due to the redistribution of loads [7]. In silico methods can help to analyze in more detail the consequences of soft tissue hardening and identify tendencies in their effect on the state of bone tissue, which cannot be tested by experimental methods. Numerical simulation techniques can help to save time and reduce the costs of investigations.

Despite their undeniable advantages, in silico methods present several challenges. Firstly, the development of geometrical models requires accurate processing of the joint's anatomical geometry, which demands appropriate software packages capable of reading medical images. Secondly, the problem formulation is another difficulty, as biological tissues generally behave as nonlinear elastic or hyperelastic bodies, requiring the use of more intricate mathematical techniques, and the material data should be well known. All of the mentioned points should be verified for sufficient investigation to be performed.

Recent studies, for the most part, suggest using the finite element method for numerical modeling of the knee. A literature review has shown that the problems and research objectives that are addressed can be very diverse: authors [8] pronounced a numerical knee model for verifying the mechanical processes in knee joint in case of arthroplasty. The geometry of the model was obtained by processing DICOM data. The three-dimensional model contains femur, tibia, ligaments, and implants, although the bone components didn't reflect the trabecular structure; in addition, the ligaments were modeled as perfectly elastic bodies, so nonlinear behavior was not verified. The model proposed by [9] allowed the verification of the loads occurring in cases of flexion and was characterized by complex external geometry: it contained fibula and reflected the anatomical curvature of bone surfaces, though the trabecular structure of porous bone and the hyperelasticity of cartilages and ligaments were not verified too. Other authors [10] provided a shock wave of research on the knee joint affected with osteoarthritis. The porous material properties were taken into account, but at the same time, the representation of spongious bone was simplified, so the proposed model cannot accurately predict the interaction between cortical and trabecular bone. Some authors [11,12] provide numerical knee joint models with verification of nonlinear properties of soft tissues, although they assume the bone components to act as rigid bodies, so the elastic properties of femur and tibia remain uninvolved.

The present study aims to investigate the stress changes of the bones which occur in cases of deterioration of cartilage. The numerical model of the knee joint containing components of femur, tibia, meniscus, and cartilage was developed. The trabecular structure of both femur and tibia was verified. In order to assess the hyperelastic properties of soft tissues, the Mooney–Rivlin material model was applied, and the problem of nonlinear theory of elasticity was raised. The importance of the work for the field of biomedical engineering is that the proposed model can be parameterized, and this, in turn, can contribute to the further study of biomechanical processes occurring in the knee joint.

## 2. Materials and Methods

### 2.1. Geometry and Structure of the Model

The three-dimensional numerical model of the knee joint was developed through several steps. At first, the computed tomography of a 64-year-old female diagnosed with third stage OA was performed. Then, the obtained images were processed using the free open-source software 3D Slicer [13] and further refined using the MeshLab program [14] to remove noise and smooth surfaces. The result of geometry enhancement in MeshLab is shown in Figure 1.

As we can see from Figure 1, initially, the model was simplified, and the number of constituent triangles was reduced from 150,304 to 25,000. This step was performed to save resources of the computing program. However, since the surface becomes angular with mesh simplification, Laplacian smoothing was performed using the same MeshLab program. Thus, the initial smoothness lost during model simplification was restored. The output STL file from MeshLab was exported to the SolidWorks software [15] environment, where the final rendering of the mesh was performed, and surfaces were transformed into solid-state models of the femur and tibia. In the same SolidWorks environment, cartilage congruent to articular surfaces was formed for bone components, assuming the same thickness (2 mm) throughout the volume. In the next stage, the meniscus and cartilage were added to the existing model components. The final numerical model is shown in Figure 2.

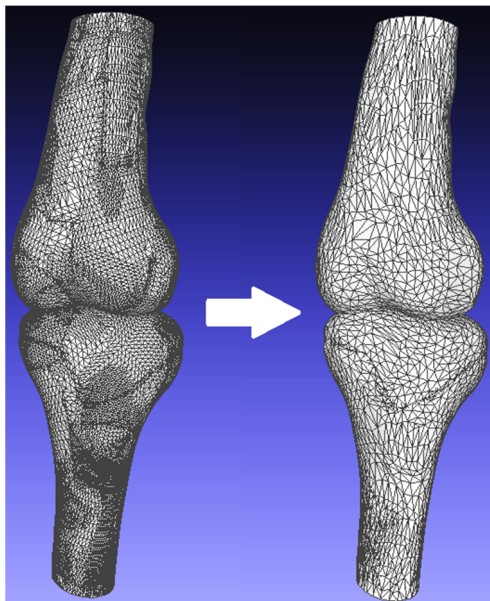

**Figure 1.** Model geometry simplification and smoothening using MeshLab.

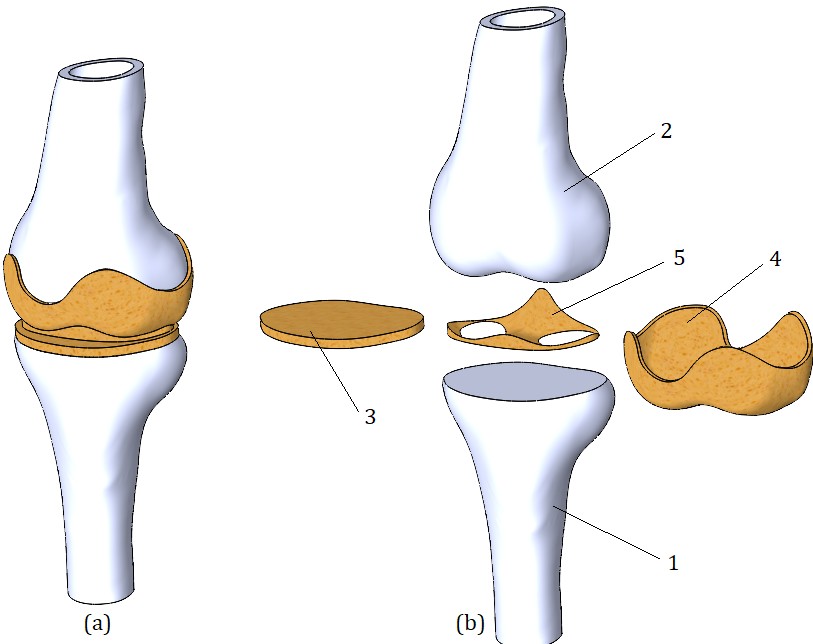

**Figure 2.** (**a**) Numerical knee joint model; (**b**) Model components: 1—tibia, 2—femur, 3—tibia cartilage, 4—femur cartilage, 5—meniscus.

It is important to note that the initial geometry underwent three processing steps, which led to distortion of initial geometric shape and volume. Although the model reflects the bone's characteristic curvature, errors in geometry are indeterminable due to the numerous refinements made to the initial file. Moreover, the model's limitations include the absence of ligaments, rendering it suitable only for compression load assessment, as calculating cases with rotation and flexion can result in unreliable outcomes.

The section view of the model is presented in Figure 3. The cortical bones of tibia and femur were modeled as thin features (thickness of both bones was set to 4.5 mm), while trabecular tissue of the tibia and femur was presented as porous structure by applying the regular spherical cuts. The diameter of cutting cuts was set to 6 mm, and the distance of

their geometrical centers was equal to 5 mm. In the result, the ratio of bone volume and total volume (BV/TV) was equal to 0.2. This value is in agreement with the BV/TV ratios reported by [16].

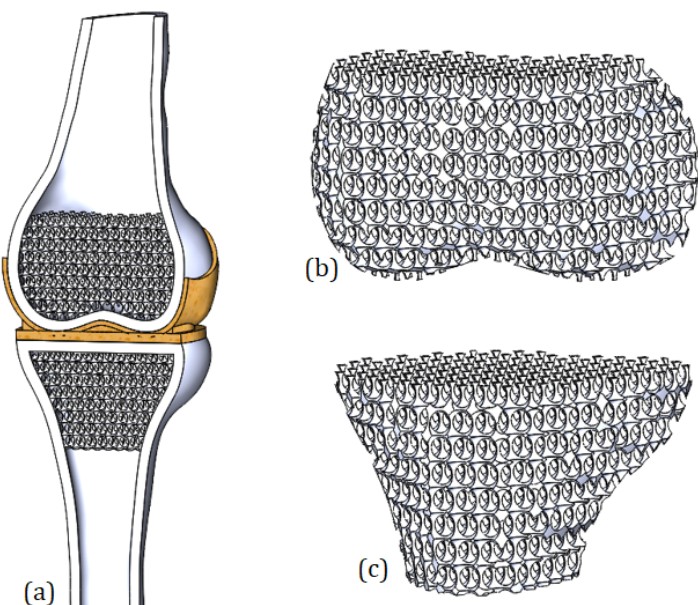

**Figure 3.** (**a**) Section view of the knee model; (**b**) Trabecular structure of femur; (**c**) Trabecular structure of tibia.

### 2.2. Mechanical Properties of Model Components

The bone tissue was assumed as a perfectly elastic continuum. The Young's modulus for bone components was set to 7.3 GPa; the Poisson's coefficient was set to 0.3 [17]. It was assumed that the critical stress of bone tissue is equal to 140 MPa [18].

The cartilage was modeled as a hyperelastic continuum. The Poisson's ratio $v$ for the cartilage and meniscus was set to 0.4995. The Mooney–Rivlin material model was applied.

The Mooney–Rivlin strain energy density function is expressed as a two-constant formulation [19]:

$$w = C_1(I_1 - 3) + C_2(I_2 - 3) + \frac{1}{2}K(I_3 - 1)^2, \tag{1}$$

where $C_1$ and $C_2$ are the first and second material constants, respectively, related to the response of distortion, and $K$ is the material constant referred to the volumetric response. $I_1$, $I_2$, and $I_3$ are the reduced invariants of the Cauchy–Green deformation tensor and can be determined in terms of principal stretch ratios.

The material constant $K$ can be expressed as:

$$K = \frac{6(C_{01} + C_{10})}{3(1 - 2v)}; \tag{2}$$

The deterioration of cartilages is modeled by varying the values of elastic constants. The more deteriorated cartilage is reflected with higher stiffness; the average level of deterioration is determined by a twice lower values of the elastic constants, while the healthiest cartilage is assigned the lowest stiffness. The range of cartilage elastic constants was reported by [20] and was involved in the study. The $C_1$ and $C_2$ values of femur and tibia are presented in Table 1.

**Table 1.** Elastic constants of femur and tibia cartilages.

| Stiffness | $C_1$, MPa (Femur) | $C_2$, MPa (Femur) | $C_1$, MPa (Tibia) | $C_2$, MPa (Tibia) |
|---|---|---|---|---|
| Mild | 0.24 | 0.03 | 0.21 | 0.04 |
| Moderate | 0.66 | 0.32 | 0.67 | 0.18 |
| Severe | 1.08 | 0.61 | 1.13 | 0.32 |

The meniscus was modeled as perfectly elastic nonlinear material. In order to determine its mechanical behavior, the stress–strain curve offered in [21] was involved in the study (Figure 4). The presented curve reflects the 3rd stage of OA.

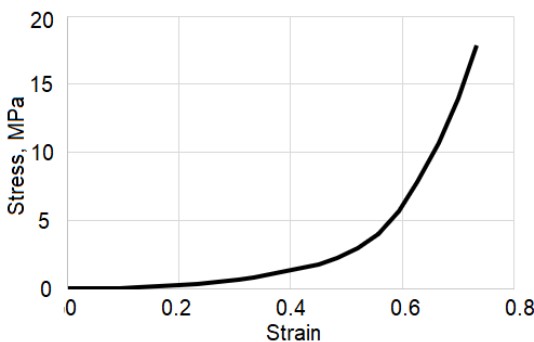

**Figure 4.** Stress–strain curve representing the mechanical properties of meniscus.

*2.3. Problem Formulation*

In order to determine the mechanical behavior of the model, the nonlinear theory of elasticity was applied [15]. In dynamic analysis, the equilibrium equations of the system at time step $t + \Delta t$ are

$$[M]^{t+\Delta t}\{U''\}^{(i)} + [C]^{t+\Delta t}\{U'\}^{(i)} + {}^{t+\Delta t}[K]^{(i)t+\Delta t}\{\Delta U\}^{(i)} = {}^{t+\Delta t}\{R\} - {}^{t+\Delta t}\{F\}^{(i-1)} \quad (3)$$

where $[M]$ is the mass matrix; $[C]$ is the damping matrix; ${}^{t+\Delta t}[K]^{(i)}$ is the stiffness matrix; ${}^{t+\Delta t}\{R\}$ is the vector of external loads; ${}^{t+\Delta t}\{F\}^{(i-1)}$ is the vector of internal forces at iteration $(i-1)$; ${}^{t+\Delta t}\{\Delta U\}^{(i)}$ is the vector of incremental displacements at iteration $(i)$; ${}^{t+\Delta t}\{U'\}^{(i)}$ is the vector of total velocities at iteration $(i)$; and $[M]^{t+\Delta t}\{U''\}^{(i)}$ is the vector of total accelerations at iteration $(i)$, where damping matrix $[C]$ was neglected, $[C] = 0$.

Employing implicit time integration Newmark–Beta scheme and using Newton's iterative method, the above equations are expressed in the form:

$$^{t+\Delta t}[K]^{(i)}\{\Delta U\}^{(i)} = {}^{t+\Delta t}\{R\}^{(i)} \quad (4)$$

where ${}^{t+\Delta t}\{R\}^{(i)}$ represents the effective load vector, and ${}^{t+\Delta t}[K]^{(i)}$ denotes the effective stiffness matrix. In order to perform calculations, the SolidWorks Simulation module is used.

In order to verify the stressed state of model components, the von Mises stress criterion was applied. It is defined in Equation (5):

$$\sigma_y = \sqrt{\frac{(\sigma_1 - \sigma_2)^2 + (\sigma_2 - \sigma_3)^2 + (\sigma_3 - \sigma_1)^2}{2}}, \quad (5)$$

where $\sigma_1$, $\sigma_2$, and $\sigma_3$ are the maximum, intermediate, and minimum principal stresses, respectively, and $\sigma_y$ is a yield stress.

### 2.4. Calculation Cases and Boundary Conditions

Two calculation cases for each cartilage stiffness types were considered, with the first case involving a 1 mm displacement, which is typical of walking, and the second case involving an increased displacement of 2 mm to simulate more extreme conditions of the knee joint during running. The values of displacements were set according to the data reported by [22,23]. Figure 5a illustrates the load schematization, where the lower surface of the tibia is fixed, and the upper surface of the femur is subjected to a vertical compression load.

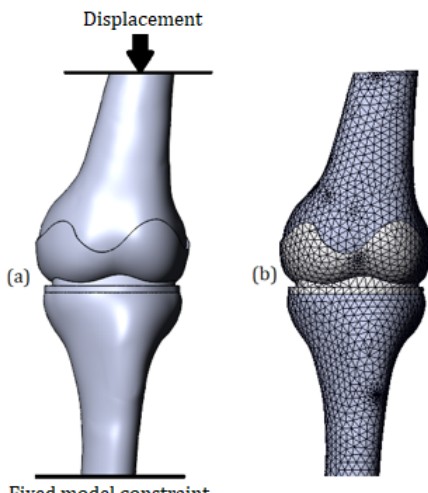

**Figure 5.** (**a**) Load schematization; (**b**) Finite element model.

The model was meshed with tetrahedral finite elements, as shown in Figure 5b. The model consisted of 191,221 finite elements and 407,378 nodes, with a maximum element size of 5.19 mm and a minimum size of 0.11 mm adapted to the trabecular bone structure. The whole model was described by 1,221,783 degrees of freedom. The equilibrium equations were solved using the Intel Direct Sparse solver.

### 3. Numerical Results and Discussion

#### 3.1. Von Mises Stress in Case of 1 mm Displacement

Von Mises stresses were calculated for the entire 3D model. A general view of the von Mises stress plot is shown in Figure 6. As can be seen from the digital scale, the maximum stress values are deployed on the bone components, while the values for soft tissues (cartilage and meniscus) are significantly lower. In the case of processing a model with mild cartilage stiffness (Figure 6a), the maximum stress values reached 11.3 MPa. For the model with moderate cartilage stiffness (Figure 6b), the maximum stress values were 12.6 MPa and were 15.4 MPa for the model with the severe cartilage stiffness (Figure 6c).

Regarding von Mises stress for soft tissues, as the cartilage stiffness increases, the stress values for femur cartilage were obtained at 0.4 MPa (Figure 6a), 1.1 MPa (Figure 6b), and 1.9 MPa (Figure 6c). The stress values for tibia cartilage were slightly higher: 0.5 MPa (Figure 6a), 1.8 MPa (Figure 6b), and 2.7 MPa. This effect can be explained by the fact that tibia cartilage typically has greater stiffness, which was taken into account. The meniscus stress values varied as follows due to changes in cartilage stiffness: 1.1 MPa (Figure 6a), 3.2 MPa (Figure 6b), and 3.9 MPa (Figure 6c). It should be noted that the stress values for the meniscus were higher than for other soft tissues. This can be explained by the fact that the meniscus was assigned properties characteristic of third-degree osteoarthritis [21].

As can be seen from Figure 6, for all analyzed cases, there is a general tendency. It lies in the fact that the most loaded nodes are located on the tubular sections of the bone. The range of stress values here varies from 9 to 11 MPa; however, we can argue that the appeared concentrators are associated more with the boundary conditions than

with the mechanical properties of the model components, since the locations of the stress concentrators are close to the location of the applied load. For this reason, for a more detailed analysis of the stress state of the model, it is necessary to study other plots, as well as section-views of the finite element model. They are presented below.

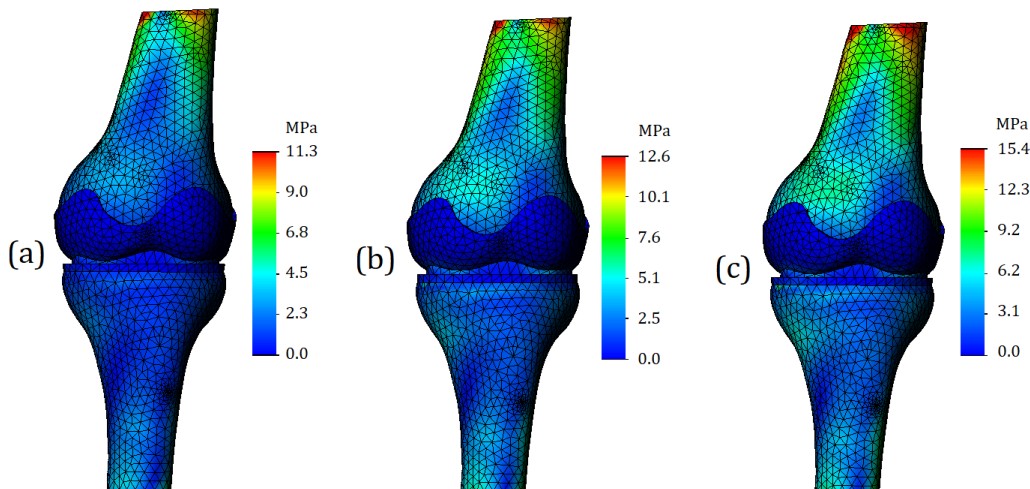

**Figure 6.** Von Mises stress plots in case of walking. General view of the model: (**a**) With mild cartilage stiffness; (**b**) With moderate cartilage stiffness; (**c**) With severe cartilage stiffness.

The finite element model section-view for the bone components is shown in Figure 7. We can observe an increase in stress values on the posterior side of both bones; however, it should be noted that this distribution is related to the geometry of the bones and the loading pattern. Firstly, the tibia and femur are not strictly collinear to each other; for this reason, the displacement that is applied to the upper surface of femur causes an external moment, due to which the model is subjected not only in compression, but also undergoes bending. As a result, we get stress concentrators at the fixed base of the model, as well as in the tubular section of femur. It must be admitted that changes in the stress state in these areas are not the direct object of our study, but, at the same time, changes in stress values in these locations allow us to conclude that the properties of the meniscus affect the load distribution not only in the case of compression but also in the case of bending.

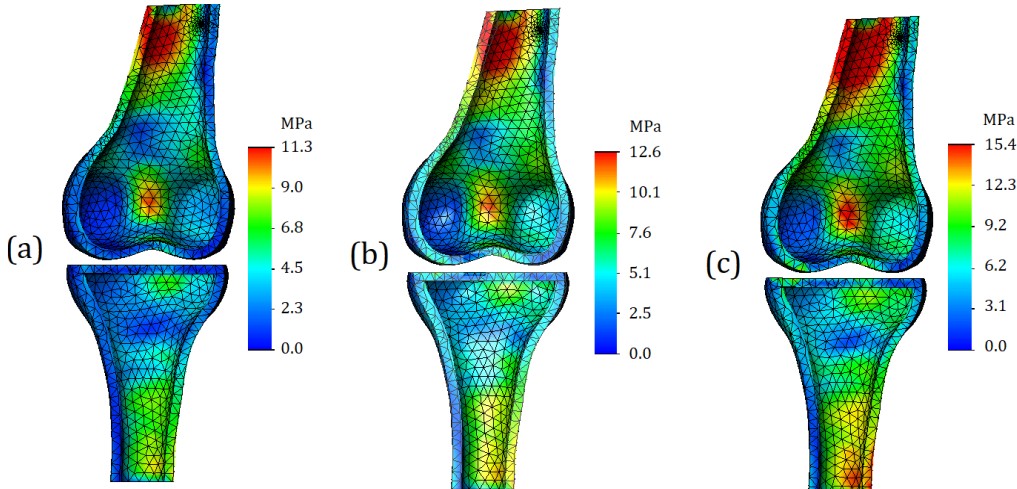

**Figure 7.** Von Mises stress plots on cortical bone section view in case of walking: (**a**) With mild cartilage stiffness; (**b**) With moderate cartilage stiffness; (**c**) With severe cartilage stiffness.

For this study, the stress in the area of the articular surfaces is of much greater interest. Plots of the stress are shown in Figure 8. It should be noted that Figure 8 depicts exactly the articular surfaces of the bone, while cartilage and soft tissues are hidden.

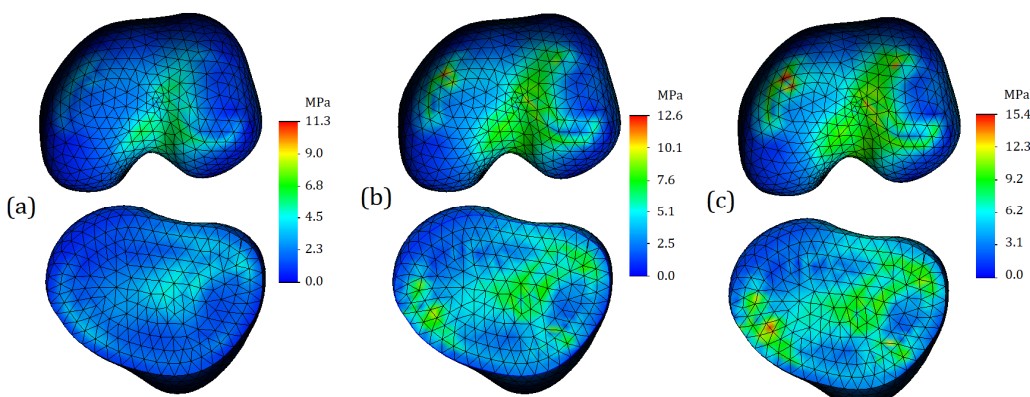

**Figure 8.** Von Mises stress plots on contact surfaces of the bones in case of walking: (**a**) With mild cartilage stiffness; (**b**) With moderate cartilage stiffness; (**c**) With severe cartilage stiffness.

As can be seen from Figure 8a, in the case of cartilage with mild stiffness, there are no stress concentrators on the articular surfaces. The zones of increased stress are not clearly expressed, and basically the model shows a uniform distribution of stresses. In the case of moderate cartilage stiffness (Figure 8b), the zones of increased stress are more clearly expressed; in addition, their area is noticeably larger. For the model with severe cartilage (Figure 8c), stress concentrators are identified in four locations (two for tibia and two for femur). However, compared to the model shown in Figure 8b, the area of the high stress remains approximately the same.

The stress plots of the trabecular tissue are shown in Figure 9. As can be seen from the plots presented, as the cartilage stiffness increases, not only do the values of the stresses themselves change, but also their distribution changes. With mild cartilage, the zones of increased stress values are more likely to be located outside, while with an increase in stiffness, stress concentrators spread deeper and cover both vertical and horizontal trabeculae. The values of the maximum stresses on the trabeculae themselves turned out to be lower than the stresses on the cortical bone: 9.6 MPa for mild stiffness cartilage, 11.2 MPa for moderate stiffness cartilage, and 12.9 MPa for severe stiffness cartilage.

### 3.2. Von Mises Stress in Case of 2 mm Displacement

Von Mises stress plots in case of running (2 mm displacement) are shown in Figure 10. In the general view of the finite element model, we can see that the "red" zones of stress values increased both in area and in numerical values. For the model with mild cartilage stiffness, the maximum stress was 14.7 MPa (Figure 10a); for a model with moderate cartilage stiffness the maximum stress was 18.5 MPa (Figure 10b); for the model with the severe cartilage stiffness the maximum stress was 3.1 MPa (Figure 10c). The model components representing cartilage, as in the previous calculation case, turned out to be painted in blue; however, the stress value in the cartilages became higher. As the stiffness increased in the model, the stress values were 1.1 MPa for the tibia cartilage and 0.8 MPa for the femur cartilage (Figure 10a), 6.7 MPa for the tibia cartilage and 1.4 MPa for the femur cartilage (Figure 10b), and 10.1 MPa for the tibia cartilage and 3.6 MPa for the femur cartilage (Figure 10c).

A significant increase in stress was also observed for the meniscus. In the case of mild cartilage stiffness, the value has risen to 4.2 MPa (Figure 10a). In the case of moderate cartilage stiffness, it was 5.1 MPa (Figure 10b), and in the case of severe cartilage stiffness—8.1 MPa (Figure 10c).

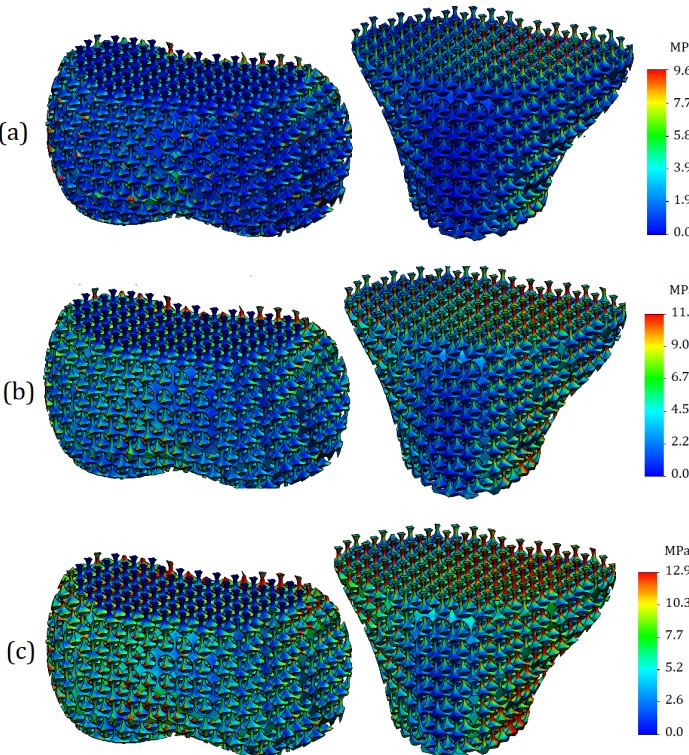

**Figure 9.** Von Mises stress plots on trabecular bone in case of walking: (**a**) With mild cartilage stiffness; (**b**) With moderate cartilage stiffness; (**c**) With severe cartilage stiffness.

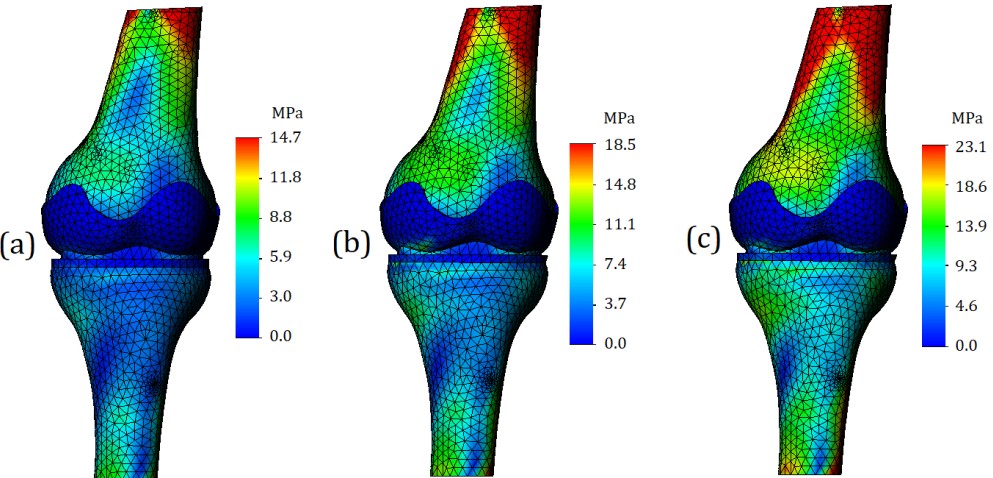

**Figure 10.** Von Mises stress plots in case of running. General view of the model: (**a**) With mild cartilage stiffness; (**b**) With moderate cartilage stiffness; (**c**) With severe cartilage stiffness.

The section view of the finite element model is shown in Figure 11. The area of zones with high stress values increased as expected. As mentioned earlier, such a distribution of stresses is caused by the external moment. However, from a strength point of view, the areas of the "red" zones (especially in the case of severe cartilage, Figure 11c), does not come close to critical stress values. Comparing the obtained values with reported yield strength of the bone (140 MPa), we find that the strength capacity under this type of loading exceeds 80 percent.

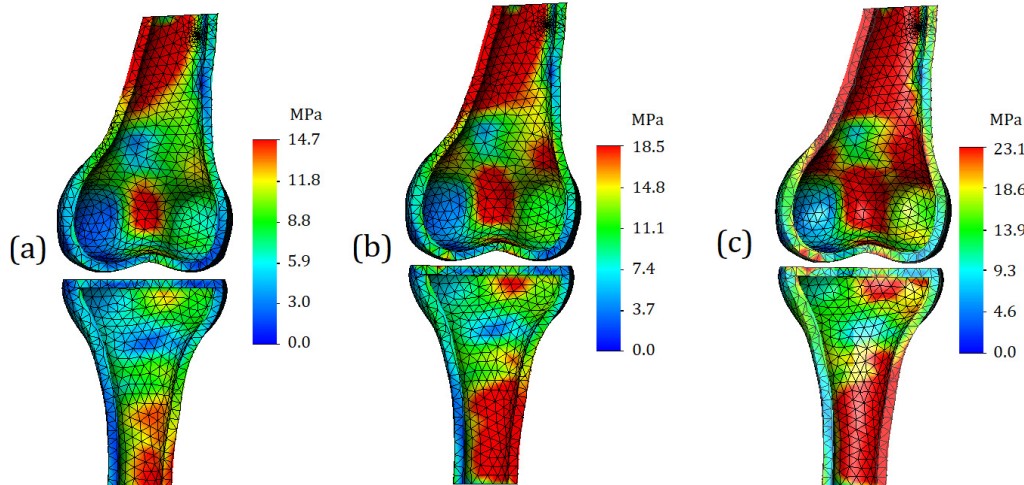

**Figure 11.** Von Mises stress plots on cortical bone section view in case of running: (**a**) With mild cartilage stiffness; (**b**) With moderate cartilage stiffness; (**c**) With severe cartilage stiffness.

The distribution of stresses on the contact surfaces of the bone is shown in Figure 12. In the case of mild cartilage (Figure 12a), small differences in values occur on the contact surface of the femur; however, they are not too intense, so it is not advisable to consider them as stress concentrators. On the contact surface of tibia, the distribution of stresses is even more uniform, which may indicate a good functionality of the cartilage: in comparison with the first case of loading, the displacement was doubled; however, this did not cause stress concentrators.

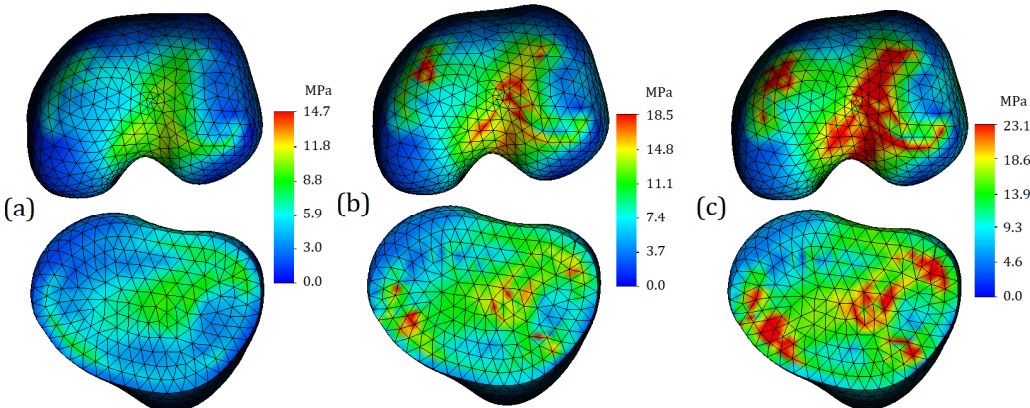

**Figure 12.** Von Mises stress plots on contact surfaces of the bones in case of running: (**a**) With mild cartilage stiffness; (**b**) With moderate cartilage stiffness; (**c**) With severe cartilage stiffness.

In the case of cartilage of moderate stiffness, with a supplied displacement of 2 mm, stress concentrators appear on both contact surfaces of the bones. They have a different area and are dispersed over the entire articular surface (Figure 12b).

In the case of severe cartilage stiffness, stress concentrators already occupy a significant part of the contact area (Figure 12c). Again, from the strength point of view, the values themselves are safe; however, such pronounced differences in values in the long term may contribute to more intense wear of tissue.

The distribution of stresses on the trabecular component of the model is shown in Figure 13. The trends described in the previous subsection are reflected, but it should be noted that in the case of higher displacement (2 mm), the resulting stress values on the trabeculae are percentage closer to the stress values on the cortical bone. The obtained values of stress are equal to 13.6 MPa, 17.4 MPa, and 22.3 MPa for calculations with mild, moderate, and severe cartilage stiffness, respectively.

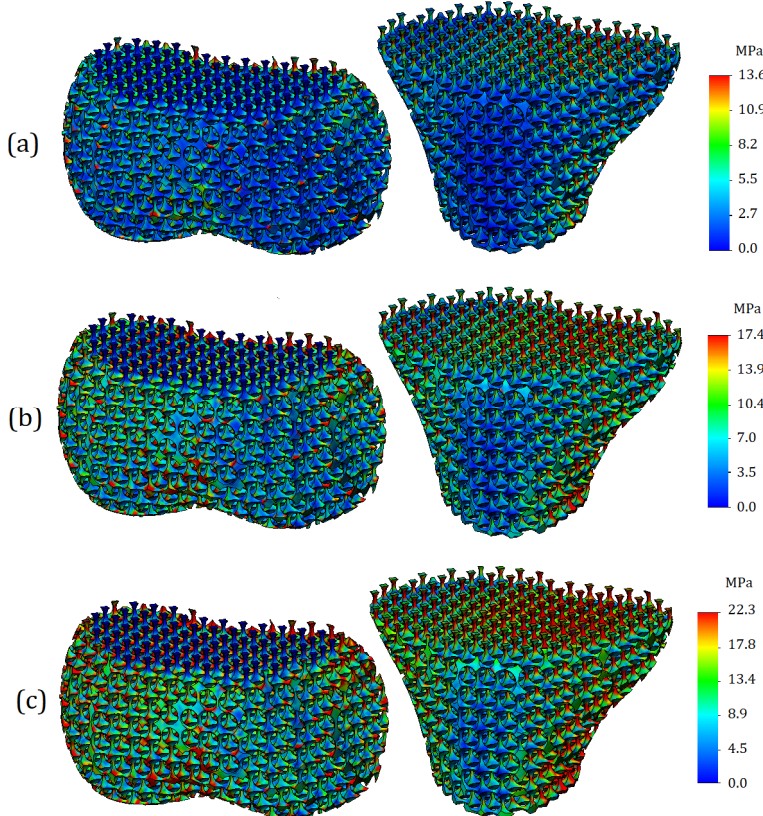

**Figure 13.** Von Mises stress plots on trabecular bone in case of running: (**a**) With mild cartilage stiffness; (**b**) With moderate cartilage stiffness; (**c**) With severe cartilage stiffness.

### 3.3. Comparison of Stress Values and Discussion

The diagram representing the obtained values of maximum stresses under different loading conditions is shown in Figure 14. As can be seen from the values, in the case of running, the stresses on bone caused by mild cartilage are lower than those in case of walking caused by moderate and severe cartilage, obtained at loading typical of normal walking.

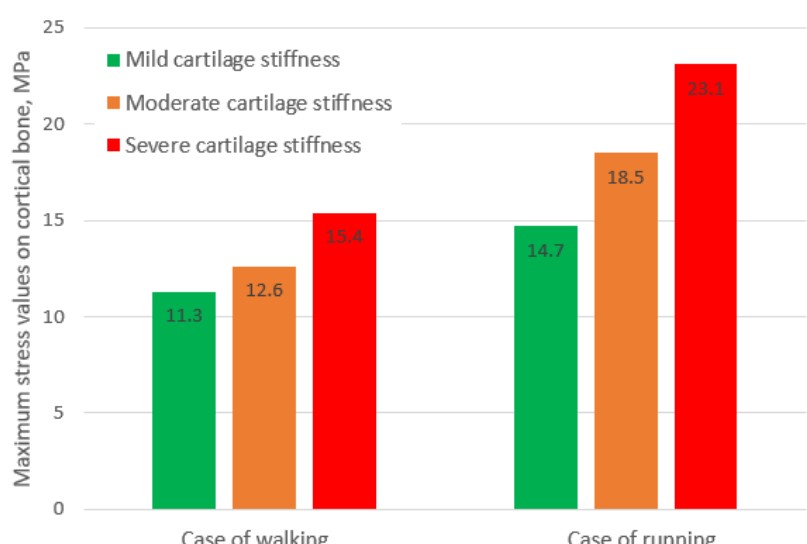

**Figure 14.** Maximum von Mises stress values on cortical bone in case of various cartilage stiffness and loading conditions.

The percentage increase in stress in cases of running is shown in Figure 15. Interestingly, in the case of soft cartilage, the growth of stress reached 31%, whereas in case of medium and high cartilage stiffness, the percentage increase in stress values was approximately the same and reached 50%.

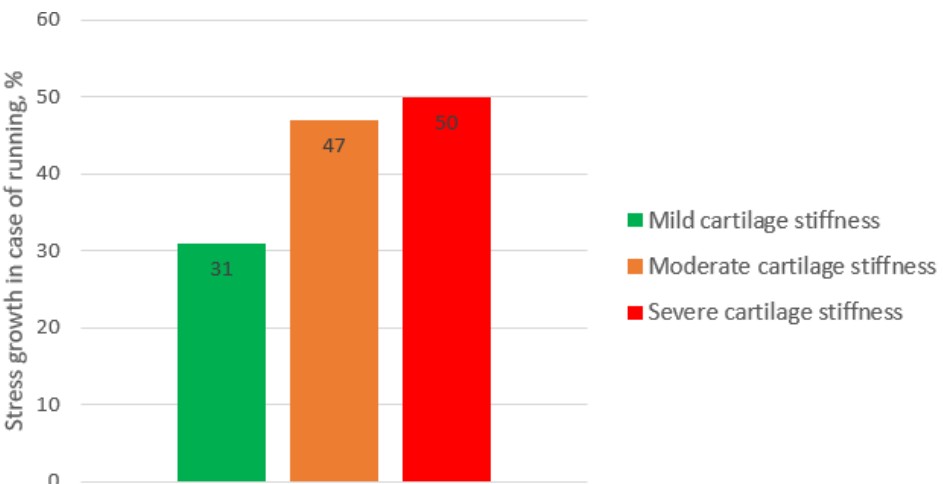

**Figure 15.** Maximum von Mises stress percentage growth in case of running.

The analysis of the obtained results allows us to state that taking into account the non-linear properties of soft tissues is extremely important for assessing the stress state of the entire biological subsystem. The main difficulty, however, is still the lack of data regarding the mechanical behavior of tissues in certain diseases. Despite the fact that general trends, such as an increase in the stiffness of cartilage in osteoarthritis, are known, modern science either does not have information on the specific values of the required constants or parameters, or this information is contradictory.

The advantages of in silico studies were emphasized in the work; however, for their full-scale application, an information base based on experimental data is necessary.

As for this specific study and the proposed methodology, its improvement can be performed in several directions. Firstly, the numerical model geometry can be improved. To more thoroughly study the processes occurring in the knee joint, the model can be supplemented with important anatomical components, such as ligaments. The presence of ligaments will allow the verification of more complex loading cases, such as flexion. Secondly, a more complex mathematical apparatus can be applied. Accounting for properties such as viscoelasticity and damping will allow for research on impact loads that the knee joint is subjected to during normal life cycles. To obtain more reliable results, it would be desirable to use more advanced material models that take into account the anisotropy of mechanical properties. The mathematical models of such materials themselves have been developed and mathematically defined, but the main difficulty is that each bone is strictly individual, and the absence of a template severely limits the application of more accurate models.

## 4. Conclusions

In this study, a numerical model was developed to assess the stress state of components within the knee joint, taking into account both the applied load and mechanical properties. The proposed novel model offers several advantages, including parameterization features that allow for the geometrical variables control, such as the porosity of trabecular tissue, the thickness of cortical bone, and the thickness of cartilage.

The performed research helped to determine some new patterns related to the non-linear behavior of soft tissues, particularly the increase in bone stress caused by the initial stiffness of cartilage, the appearance of stress concentrators on articular surfaces of the bone, and the dislocation tendencies of stress on spongious bone trabeculae. For a more thorough

implementation of the model, as well as the proposed method in medical practice, extensive experimental studies of biological samples are required, the aim of which would be to compare the mechanical properties of knee joint components to the properties inherent in degenerative diseases.

**Author Contributions:** Conceptualization, V.A., O.A. and J.S.; methodology, V.A., O.A. and J.S.; software, O.A.; formal analysis, V.A., O.A., J.S. and O.C.; investigation, V.A., O.A., J.S. and O.C.; resources, V.A.; data curation, V.A.; writing—original draft preparation, O.A., J.S. and O.C.; writing—review and editing, V.A.; visualization, O.A.; supervision, V.A.; project administration, V.A.; funding acquisition, V.A. All authors have read and agreed to the published version of the manuscript.

**Funding:** This work received no funding.

**Institutional Review Board Statement:** Not applicable.

**Informed Consent Statement:** Not applicable.

**Data Availability Statement:** The data presented in this study are available on request from the corresponding author.

**Conflicts of Interest:** The authors declare no conflict of interest.

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
