# Peer review of "In Silico Modeling the Impact of Cartilage Stiffness on Bone Tissue Stress"

_applsci, doi:10.3390/app13074457_

Round 1

Reviewer 1 Report

Current manuscript entitled “In-silico modeling the impact of cartilage stiffness on bone tissue stress” deliberated on the stress state alteration in bone tissue based on various mechanical properties of cartilage. The results demonstrated that as the stiffness of the cartilage increased, the distribution of stresses in the bone tissue became uneven and stress concentrators dispersed over articular surface, while in case of mild cartilage stiffness no stress concentrators were expressed. Manuscript can be accepted after addressing the following comments.

1.      Abstract:

- Please include the problem statement in the abstract to show why is this study important

- Include some discussions that are discussed in the paper

- The abstract is not properly wrapped up, please revise

2.      Authors can improve the introduction section by discussing the relevant and recent studies. Also, please mention the important of this study to society as well as industry.

3.      Check for the grammatical errors in the manuscript.

4.      Clear statements of novelty should appear in the conclusions as well.

5.      Please eliminate the use of redundant words.

Author Response

Dear Reviewer, thank You for reviewing our manuscript. We’ve tried our best to improve it according to Your recommendations.

Point 1: Abstract:

- Please include the problem statement in the abstract to show why is this study important

- Include some discussions that are discussed in the paper

- The abstract is not properly wrapped up, please revise

 Response 1: The problem statement in abstract is included, the importance of work is expressed. The discussions are added and the abstract is revised in purpose to make it in more appropriate way.

Point 2: Authors can improve the introduction section by discussing the relevant and recent studies. Also, please mention the important of this study to society as well as industry.

Response 2: The introduction section is improved, the discussion of relevant works is added. The importance of work for society and industry is mentioned.

Point 3: Check for the grammatical errors in the manuscript.

Response 3: The English in the paper is improved.

Point 4: Clear statements of novelty should appear in the conclusions as well.

Response 4: Novelty statements are added to conclusions.

Point 5: Please eliminate the use of redundant words.

Response 5: The use of redundant words is eliminated.

Reviewer 2 Report

The Article: Title "In-silico modeling the impact of cartilage stiffness on bone tissue stress" is about examine the stress state alteration in bone tissue based on various mechanical properties of cartilage. The results showed that as the stiffness of the cartilage increased, the distribution of stresses in the bone tissue became uneven and stress concentrators dispersed over articular surface, while in case of mild cartilage stiffness no stress concentrators were expressed. The main question addressed by the research is clear . The topic is original  and it add new information. It seems that no specific improvements should the authors consider regarding the methodology. The conclusions consistent with the evidence and arguments and  they address the main question. The references are appropriate. The tables and figures clear ilustrated the topic.

Author Response

Point 1: The Article: Title "In-silico modeling the impact of cartilage stiffness on bone tissue stress" is about examine the stress state alteration in bone tissue based on various mechanical properties of cartilage. The results showed that as the stiffness of the cartilage increased, the distribution of stresses in the bone tissue became uneven and stress concentrators dispersed over articular surface, while in case of mild cartilage stiffness no stress concentrators were expressed. The main question addressed by the research is clear. The topic is original and it add new information. It seems that no specific improvements should the authors consider regarding the methodology. The conclusions consistent with the evidence and arguments and they address the main question. The references are appropriate. The tables and figures clear ilustrated the topic.

 Response 1: Dear Reviewer, thank You for you kind response. It is very inspiring, and we will try our best to make more.

Reviewer 3 Report

The authors developed a numerical model to assess the stress state of components within the knee joint. It is a valuable research work. However, there is some questions in the manuscript. After considering the following comments, I suggest that this manuscript should be published.

1. Figure 14 and 15 is not very clear, please change them.

2. The advantages and disadvantages of in-silico methods should be discussed in detail in part Introduction.

3. The matching mark in equation (3) should be consistent with the explanation below.

Author Response

Dear Reviewer, Thank You for reviewing our manuscript. We agree with every note and improved our manuscript according to Your recommendations.

Point 1: Figure 14 and 15 is not very clear, please change them.

Response 1: Figures 14 and 15 are improved and the clarifying data are added.

Point 2: The advantages and disadvantages of in-silico methods should be discussed in detail in part Introduction.

Response 2: The discussion on advantages and disadvantages of in-silico methods to the Introduction section are added.

Point 3: The matching mark in equation (3) should be consistent with the explanation below.

Response 3: The explanation of equation is corrected.

Reviewer 4 Report

prior to acceptance, I would like to recommend a thorough English language edition, as some sentences are awkward, which may hamper readability and correct grasping of the content. E.g. the sentence in l. 261-264 should rather be: "As mentioned earlier, such a distribution of stresses is caused by the external moment that has arisen, however, from a strength point, the vast areas of the "red" zones (especially in the case of hard cartilage, 11c), do not represent the critical stress values."

Author Response

Point 1: Prior to acceptance, I would like to recommend a thorough English language edition, as some sentences are awkward, which may hamper readability and correct grasping of the content. E.g. the sentence in l. 261-264 should rather be: "As mentioned earlier, such a distribution of stresses is caused by the external moment that has arisen, however, from a strength point, the vast areas of the "red" zones (especially in the case of hard cartilage, 11c), do not represent the critical stress values."

Response 1: Dear Reviewer, thank You for reviewing our manuscript. We totally agree with your suggestions, so the English in our work is now improved.
